

**PeerJ Hubs**
Published on behalf of

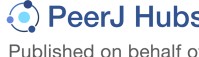
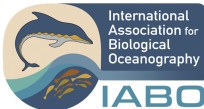
International Association for Biological Oceanography
IABO

# Viral metagenomic investigation of two Caribbean echinoderms, *Diadema antillarum* (Echinoidea) and *Holothuria floridana* (Holothuria)

Ian Hewson[1], Marilyn Brandt[2], Kayla Budd[2], Mya Breitbart[3], Christopher DeRito[1], Samuel Gittens Jr[2], Michael W. Henson[4], Alwin Hylkema[5,6], Moriah Sevier[2], Matthew Souza[2], Brayan Vilanova-Cuevas[1] and Sarah Von Hoene[2]

[1] Cornell University, Ithaca, NY, United States of America
[2] University of the Virgin Islands, St Thomas, US Virgin Islands, Virgin Islands
[3] University of South Florida, St Petersburg, FL, United States of America
[4] Northern Illinois University, DeKalb, IL, United States of America
[5] Wageningen University and Research, Wageningen, Netherlands
[6] Van Hall Larenstein University of Applied Sciences, Leeuwarden, Netherlands

Corresponding author
Ian Hewson, hewson@cornell.edu

## ABSTRACT

**Background**. Echinoderms play crucial roles in coral reef ecosystems, where they are significant detritivores and herbivores. The phylum is widely known for its boom and bust cycles, driven by food availability, predation pressure and mass mortalities. Hence, surveillance of potential pathogens and associates of grossly normal specimens is important to understanding their roles in ecology and mass mortality.

**Methods**. We performed viral surveillance in two common coral reef echinoderms, *Diadema antillarum* and *Holothuria floridana*, using metagenomics. Urchin specimens were obtained during the 2022 *Diadema antillarum* scuticociliatosis mass mortality event from the Caribbean and grossly normal *H. floridana* specimens from a reef in Florida. Viral metagenomes were assembled and aligned against viral genomes and protein encoding regions. Metagenomic reads and previously sequenced transcriptomes were further investigated for putative viral elements by Kraken2.

**Results**. *D. antillarum* was devoid of viruses typically seen in echinoderms, but *H. floridana* yielded viral taxa similar to those found in other sea cucumbers, including *Pisoniviricetes* (Picornaviruses), *Ellioviricetes* (Bunyaviruses), and *Magsaviricetes* (Nodaviruses). The lack of viruses detected in *D. antillarum* may be due to the large amount of host DNA in viral metagenomes, or because viruses are less abundant in *D. antillarum* tissues when compared to *H. floridana* tissues. Our results also suggest that RNA amplification approach may influence viral representation in viral metagenomes. While our survey was successful in describing viruses associated with both echinoderms, our results indicate that viruses are less pronounced in *D. antillarum* than in other echinoderms. These results are important in context of wider investigation on the association between viruses and *D. antillarum* mass mortalities, since the conventional method used in this study was unsuccessful.

## INTRODUCTION

Echinoderms play significant ecological roles in coral reef ecosystems as herbivores and detritivores (*Birkeland, 1989*). In coral reef ecosystems, echinoids, which are one group of echinoderms, contribute to the balance between macroalgae and coral cover by removing algae and opening space for coral recruits (*Edmunds & Carpenter, 2001*) and reducing dissolved organic matter inputs, disfavoring coral pathogens (*Haas et al., 2016*; *Nelson, Wegley Kelley & Haas, 2023*). Another group of echinoderms, holothurians, contribute to biogeochemical cycling through detritivory and deposit feeding, influencing remineralization of sediment-bound organic material (*Uthicke, 2001a*; *Uthicke, 2001b*; *Williamson et al., 2021*). Both echinoids and holothurians face considerable threats to their population densities through disease (*Lessios et al., 1984*; *Delroisse et al., 2020*) and harvesting (*Purcell et al., 2013*; *Wolfe & Byrne, 2022*), which may alter coral reef ecosystem structures in the future. Hence, there is a great need to understand factors potentially influencing their mortality, including identifying potential pathogens and other stressors that may contribute to host-microbe interactions.

The long-spined sea urchin (*Diadema antillarum*) is a crucial constituent of Caribbean reefs that has experienced two distinct mass mortality events in the past 50 years. Between 1983 and 1984, around 98% of *D. antillarum* across the Caribbean died (*Lessios, 1988*; *Lessios, 1995*; *Lessios, 2016*), which contributed to reef degradation through alleviated herbivore pressure on macroalgae. Macroalgae compete with corals for space, light and nutrients and their proliferation severely inhibited coral recruitment (*Mumby, Hastings & Edwards, 2007*). Unfortunately, the etiology of the early 1980s die-off was never determined, and no abnormal specimens exist today. In early 2022, another mass mortality event affected *D. antillarum*, beginning on the western side of St. Thomas (U.S. Virgin Islands) and progressing across most of the windward islands before dissipating in late summer and fall 2023 (*Hylkema et al., 2023*). The 2022 mass mortality was caused by a scuticociliate most closely related to *Philaster apodigitiformis* (*Vilanova-Cuevas et al., 2023*). This pathogen colonized spines, body wall, tube feet, and eventually the coelomic cavity of affected specimens (*Hewson et al., 2023*). This highly deleterious pathogen was first identified through host tissue transcriptomics (*Hewson et al., 2023*).

The sea cucumber *Holothuria floridana* (cf *mexicana*) inhabits Caribbean coral reefs. This species is amongst the most abundant holothurians on reef flats (*Guzmán & Guevara, 2002*), where individuals are less mobile than predicted by random walk models, suggesting they are constrained by habitat heterogeneity (*Hammond, 1982*). *H. floridana* consumes more sediment than sympatric species and contributes to nitrogen cycling through release of ammonium at rates higher than co-occurring holothurian taxa (*Munger et al., 2023*). Its fecal pellets may persist for longer than other species (*Conde, Díaz & Adriana, 1991*). As with most coastal holothurians, *H. floridana* hosts a diverse assemblage of epibionts, including crustacea, annelids, and mollusks (*Rogers et al., 2024*). The virome composition

of *H. floridana* has not been previously assessed. However, holothurians generally are known to harbor diverse viruses (*Hewson, Johnson & Tibbetts, 2020*; *Jackson et al., 2022*; *Wang et al., 2021a*; *Wang et al., 2021b*), some of which may be associated with disease processes (*e.g.*, skin ulceration disease and visceral ejection syndrome; *Deng et al., 2008*). Most viral metagenomic approaches have not been followed by study of their pathology, and as such, it is not possible to assess their roles in holothurian health. Recently, we described a novel flavivirus infecting *Apostichopus californicus*, and anticipated that these may be common amongst sea cucumbers (*Hewson, Johnson & Tibbetts, 2020*). However, surveys across habitats and between species revealed a tight association between this virus and its host, and the virus was only found in the geographic area where it was originally observed. This suggests holothurian viruses may have restricted ranges (*Crandell et al., 2023*). Furthermore, the flavivirus did not appear related to host mass or tissue protein content, and only weakly and positively with tissue lipid content, so its role in host health remains unclear (*Crandell et al., 2023*).

Viruses infect every phylum of life and play crucial roles in marine food webs by releasing dissolved organic materials, regulating population densities of dominant resource competitors, and facilitating gene exchange (*Breitbart, 2012*; *Suttle, 2005*; *Suttle, 2007*; *Wilhelm & Suttle, 1999*). Viral metagenomics has elucidated an astonishing array of viral diversity both as free agents in seawater and sediments, and in association with hosts (*Breitbart et al., 2004*; *Breitbart et al., 2002*; *Jackson et al., 2022*). In addition, mining of invertebrate transcriptomes has expanded the host range of many viral taxa and allowed identification of putative pathogenic agents (*Shi et al., 2016*). Most viruses identified by genomic surveys of animal tissues are not believed to be pathogenic in these hosts, yet investigations provide important information for downstream autecological study (*Gudenkauf et al., 2014*; *Hewson et al., 2014*).

This study surveilled viruses of *H. floridana* and *D. antillarum* as part of a wider effort to understand the roles of their associated microbiomes in animal health. The impetus for this survey was to identify candidate viral agents which could be used, downstream, in quantitative assessment of their association with tissue health. We surveyed viral metagenomes of *D. antillarum* that were grossly normal and those affected by *D. antillarum* scuticociliatosis (DaSc) collected from St. Thomas, St. John (U.S. Virgin Islands) and Saba (Caribbean Netherlands) at reference and DaSc-affected sites. We also surveyed viral metagenomes from grossly normal *H. floridana* near Marathon, Florida. Both species were targeted because of their important roles in coral reef ecology.

## MATERIALS & METHODS

### Sample collection

*D. antillarum* were collected from three locations during the boreal spring 2022 *D. antillarum* scuticociliatosis (DaSc) outbreak from St. Thomas and St. John, U.S. Virgin Islands and from Saba, Caribbean Netherlands as part of a study to examine disease etiology (*Hewson et al., 2023*). Whole animal specimens were collected in Spring 2022 by snorkel (Brewers Bay, St. Thomas [18.3407N, 64.9769W; 16 February 2022] and Tide Pools, Saba

[17.6433N, 63.2186W; 7 April 2022]) or scuba divers (Pope Point [18.3451N, 64.6938W; 21 April 2022] and Long Point [18.3324N, 64.6786W; 21 April 2022]), St. John, and Diadema City, Saba [17.6147N, 63.2489W; 7 April 2022]), and brought to the surface in individually sealed gallon plastic bags (St. John) or buckets (Saba) for transport to facilities for storage or dissection, *via* shore access or a small support vessel and vehicle. Within 3 hours of collection all specimens were frozen or dissected. Samples from dissections (body wall, digestive tract and gonad fractions) were immediately preserved in liquid nitrogen. Specimens were transported to the laboratory at Cornell University for further processing. Specimens ($n = 5$) of grossly normal *H. floridana* were collected offshore of Marathon, Florida (24.7592N, 81.1076W), in August 2023 and immediately subsampled by five mm biopsy punches, which were transferred to sterile cryovials per *Crandell et al. (2023)*. Punches were frozen at $-80\ °C$ on arrival to the lab at the Florida Fish and Wildlife Commission (within an hour of collection) before shipping to Cornell University. Samples of *D. antillarum* were thawed in the lab, then small sections of body wall from the aboral surface were removed for further analysis. *D. antillarum* were collected from St. Thomas under permit DFW22033U and St. John under permit VICR-2022-SCI-0007, and in Saba under auspices of the Saba Conservation Foundation, management authority of the Saba Marine Park. Specimens of *Holothuria floridana* were collected under permit FKNMS-2023-057 issued to the Florida Fish and Wildlife Conservation Commission (W. Sharp).

## RNA viral metagenome library preparation

Small subsamples of each specimen were prepared for viral metagenomic sequencing broadly following the approach of *Ng et al. (2011)*. Briefly, tissues were homogenized in sterile, $0.02\ \mu m$ filtered phosphate buffered saline by beating using Zymo ZR Basher Beads (part number S6012-50) in a Biospec Instruments homogenizer for 1 min at maximum speed. The homogenates were centrifuged at $3,000 \times g$ for 1 min to remove large cell debris, before filtering the supernatant through $0.2\ \mu m$ pore size polyethersulfone syringe filters. Filtered homogenates were treated with 5U Turbo DNase (Invitrogen; part number AM2238), 5U Benzonase nuclease (Sigma Aldrich; part number E1014) and 20U RNase ONE (Promega; part number M4261) for 2 hours at $37\ °C$ to digest extracellular and non-ribosome-bound nucleic acids. Following incubation, RNA was extracted from subsamples of purified virus-sized material using the Zymo Viral RNA kit (part number R1034) before storing RNA at $-80\ °C$ until further processing.

Processing of extracted nucleic acids followed two protocols (Table 1). Initially, RNA in viral extracts was amplified using the SeqPlex RNA kit (Sigma Aldrich; part number SEQXE-10RXN, $n = 11$ libraries). Subsequent specimens were amplified using the TransPlex kit (WTA2; Sigma Aldrich; part number WTA2-10RXN, $n = 3$ libraries), which was used in prior RNA viral metagenomic work. Amplification products were purified using the Zymo DNA Clean and Concentrator -5 (part number D4004), quantified by Pico Green fluorescence (Invitrogen; part number P11496), and submitted for sequencing at the Cornell Biotechnology Resource Center. Libraries were prepared for sequencing using the Nextera Flex kit (Illumina; part number 20018704) and sequenced on the Illumina MiSeq

platform using the 500 bp Nano kit. Sequences from all libraries were archived at NCBI under Bioproject PRJNA1117494 and SRA accessions SRR29258987–SRR29259000.

## Bioinformatic analyses

Analysis of all libraries was initially performed in Galaxy (*The Galaxy Community, 2022*). Sequence libraries were first trimmed for adapters and low-quality sequences using Trimmomatic using default settings (paired-end, no initial ILLUMINACLIP step, sliding window timming, four bases to average across, average quality of 20), and the quality of the remaining data was assessed by FASTQC (Length of k-mer to look for = 7; *Lo & Chain, 2014*). To eliminate reads associated with host tissues, libraries were compared to host genomes using Bowtie2 (default settings only; *Langmead & Salzberg, 2012*) against the *Diadema antillarum* (NCBI accession GCA_030407125.1; *Majeske et al., 2022*) and *Holothuria leucospolita* (NCBI accession GCA_029531755.1; *Chen et al., 2023*) genomes as appropriate. Assembly followed two protocols separately for each of the two hosts. First, trimmed and host depopulated libraries were individually assembled using the Trinity algorithm (default settings; *Grabherr et al., 2011*). Libraries were also combined and globally assembled using MetaSPAdes (auto k-mer detection, auto Phred quality offset; *Nurk et al., 2017*).

## Contig spectra annotation

Contiguous sequences (contigs) from both individual and global assemblies were compared against several databases to identify putative viral fragments. Spectra were compared against all RNA viruses (*i.e., Riboviria*), unclassified viruses, and DNA viruses (*Monodnaviria*, and *Duplodnaviria*) proteins (downloaded on 2/15/24 from NCBI Viral Genomes) by BLASTx (*Altschul et al., 1990*) using an $e$-value cutoff of $10^{-20}$. We elected these genome databases since even near-identical matches across other DNA viral groups (*e.g., Varidnaviria*) may yield spurious annotations to host genomic elements or microorganisms that share homology with viral genome elements. Aligned sequences were then subject to BLASTn and BLASTx against the non-redundant database at NCBI, where any matches to cellular organisms at an $e$-value of $<1e^{-20}$ were removed from further analysis. Remaining sequences are referred to as "viral contigs" throughout this report. Viral contigs have been archived at NCBI under accession numbers PP872373–PP872393.

Viral contigs from *D. antillarum* were further analyzed by comparison against transcriptomes prepared from this taxon during study of *D. antillarum* scuticociliatosis (DaSc; *Hewson et al., 2023*) by BLASTn at an $e$-value cutoff of $1e^{-20}$. Finally, viral contigs matching *Riboviria* proteins were further compared to the transcriptome shotgun assembly (TSA) archive at NCBI by tBLASTx at an $e$-value cutoff of $1e^{-20}$.

## Viral metagenome and transcriptome read annotation

Reads in individual *D. antillarum* viral metagenomes and transcriptomes prepared as part of previous work (*Hewson et al., 2023*) (Table S1) were further studied using Kraken2 (confidence 0, minimum base quality 0, minimum hit groups 2; *Lu & Salzberg, 2020*) against the Viruses (2019) database. Annotations were taxonomically assigned according to NCBI Viral Genomes.

Hewson et al. (2024), *PeerJ*, DOI 10.7717/peerj.18321

**Table 1** Library and assembly data for viral metagenomes prepared from *Diadema antillarum* (D) and *Holothuria floridana* (H).

| Library | RNA prep | Untrimmed reads | Trimmed reads | Non-Host reads | Host reads | Host match % | # Contigs | Viral contigs | Species | Tissue type | Animal condition | Site | Collection date |
|---|---|---|---|---|---|---|---|---|---|---|---|---|---|
| DaDis | S | 336,561 | 254,176 | 31,654 | 222,522 | 87.54 | 2,930 | 0 | *D* | BW | U | STT | 2/16/22 |
| DaHealth | S | 397,901 | 287,500 | 35,223 | 252,277 | 87.74 | 1,260 | 0 | *D* | BW | H | STT | 2/16/22 |
| DaDisVir | T | 4,143,700 | 2,643,903 | 617,670 | 2,026,233 | 76.64 | 43,623 | 3 | *D* | BW | U | STT | 2/16/22 |
| DaHealthVir | T | 10,506,365 | 7,029,469 | 1,283,683 | 5,745,786 | 81.74 | 5,051 | 0 | *D* | BW | H | STT | 2/16/22 |
| vDaBW13_A | S | 202,389 | 163,828 | 18,125 | 145,703 | 88.94 | 2,096 | 0 | *D* | BW | H | STJ | 4/21/22 |
| vDaBW17_A | S | 2,755,315 | 1,995,458 | 399,631 | 1,595,827 | 79.97 | 5,542 | 0 | *D* | BW | U | STJ | 4/22/22 |
| vDaI_13 | S | 1,922,523 | 1,503,799 | 168,835 | 1,334,964 | 88.77 | 22,263 | 0 | *D* | DT | H | STJ | 4/21/22 |
| vDaI_17 | S | 3,299,795 | 2,194,708 | 678,002 | 1,516,706 | 69.11 | 6,863 | 0 | *D* | DT | U | STJ | 4/22/22 |
| vDaG_17 | S | 1,872,745 | 1,488,669 | 149,455 | 1,339,214 | 89.96 | 9,899 | 0 | *D* | G | U | STJ | 4/22/22 |
| vDaG_13 | S | 1,725,702 | 1,280,945 | 237,130 | 1,043,815 | 81.49 | 9,460 | 0 | *D* | G | H | STJ | 4/21/22 |
| Sample 1 | S | 2,739,289 | 2,371,561 | 240,796 | 2,130,765 | 89.85 | 72,188 | 117 | *H* | BW | H | FLK | 8/25/23 |
| Sample 3 | S | 3,174,424 | 2,741,167 | 282,371 | 2,458,796 | 89.70 | 72,601 | 63 | *H* | BW | H | FLK | 8/25/23 |
| Sample 4 | S | 7,831,938 | 6,755,422 | 769,111 | 5,986,311 | 88.61 | 58,741 | 3 | *H* | BW | H | FLK | 8/25/23 |
| FKSCJan | T | 1,472,143 | 1,069,805 | 196,247 | 873,558 | 81.66 | 23,237 | 31 | *H* | BW | H | FLK | 8/25/23 |
| Global Ass. | – | 15,217,794 | 12,937,955 | 1,488,525 | 11,449,430 | 88.49 | 238,167 | 199 | *D* | n/a | n/a | n/a | n/a |
| Global Ass. | – | 27,162,996 | 18,842,455 | 3,619,408 | 15,223,047 | 80.79 | 814,902 | 6 | *H* | BW | H | FLK | 8/25/23 |

**Notes.**

BW, Body wall; CF, Coelomic fluid; DT, Digestive tract; G, Gonad.

Animal Condition (U, DaSc Affected; H, Grossly Normal). Collection Site STT, St. Thomas, STJ, St. John, FLK, Marathon, Florida. Libraries were prepared for sequencing (RNA Prep) using two approaches, the TransPlex WTA2 kit (T) and the SeqPlex kit (S), both from Sigma-Aldrich.

## Phylogenetic reconstruction

Phylogenetic reconstruction was performed on translated amino acid sequences of most *Picornavirales* hits in MEGA X, including close relatives in the nonredundant (nr) database at NCBI and TSA hits (*Kumar et al., 2018*). Amino acid sequences were aligned with MUSCLE (*Edgar, 2004*) then trimmed to overlapping regions. The best amino acid substitution and distribution models were identified based on maximum likelihood in MEGA X, then phylogenetic representations were created using the nearest-neighbor interchange heuristic model per previous work on echinoderm viral metagenomic effort (*Hewson, Johnson & Tibbetts, 2020*).

# RESULTS

## Viral metagenome characteristics

Sequencing of RNA amplicons from purified viral material resulted in 42,380,790 reads for *D. antillarum* and *H. floridana*, combined, of which 63% passed Trimmomatic trimming (Table 1). Of these, 5,107,933 (19%) did not match host genomes *via* Bowtie2, where 69.1–90.0% of reads from individual libraries matched hosts. Non-host reads assembled into a total of 108,987 contigs in individual libraries for *D. antillarum*, 226,767 contigs in individual libraries of *H. floridana*. Global assemblies yielded 238,167 contigs for *D. antillarum* and 814,902 contigs for *H. floridana*. The large proportion of host DNA in each viral metagenome is consistent with previous works (*Hewson, Aquino & De Rito, 2020*; *Jackson et al., 2020*) suggesting that strategies to remove host nucleic acids from purified viruses are only partially successful.

## *Diadema antillarum* viral metagenomes and viral surveillance of transcriptomes

BLASTn/BLASTx comparison of the *D. antillarum* contigs against viral genomes and proteins yielded only three contigs from individual libraries (all matching the microvirus Sinsheimervirus phiX174, which is used as an internal spike during library preparation and therefore a contaminant), whereas in the global assembly, five contigs matched *Caudoviricetes* (42–100% similarity to sequences in Genbank by BLASTn) and one contig matched a totivirus infecting the fungus *Malassezia restricta* (100% similarity). Annotation of *D. antillarum* viral metagenomic reads and host transcriptomes (*Hewson et al., 2023*) *via* Kraken2 revealed between 0.105–0.646% of viral metagenomes and 0.037–2.993% of transcriptome reads matching viral genomes. Amongst these matches were well-represented bacteriophage, archaeal viruses, and unclassified viruses, including both dsDNA and ssDNA viruses (Fig. 1). Retroviruses were also well represented in both viral metagenomes and transcriptomes but may represent endogenized retroviruses or expressed reverse transcriptases that are homologous between viruses and cellular organisms. Amongst RNA viruses, *Pisoniviricetes* (positive sense ssRNA viruses, including picornaviruses) were well represented in transcriptomes, but were less well represented in viral metagenomes. Furthermore, *Ellioviricetes* (negative sense ssRNA viruses, including bunyaviruses) were also more prevalent in both transcriptomes and viral metagenomes than other viral families. The detection of reads matching RNA viruses, but not contigs, suggests that the
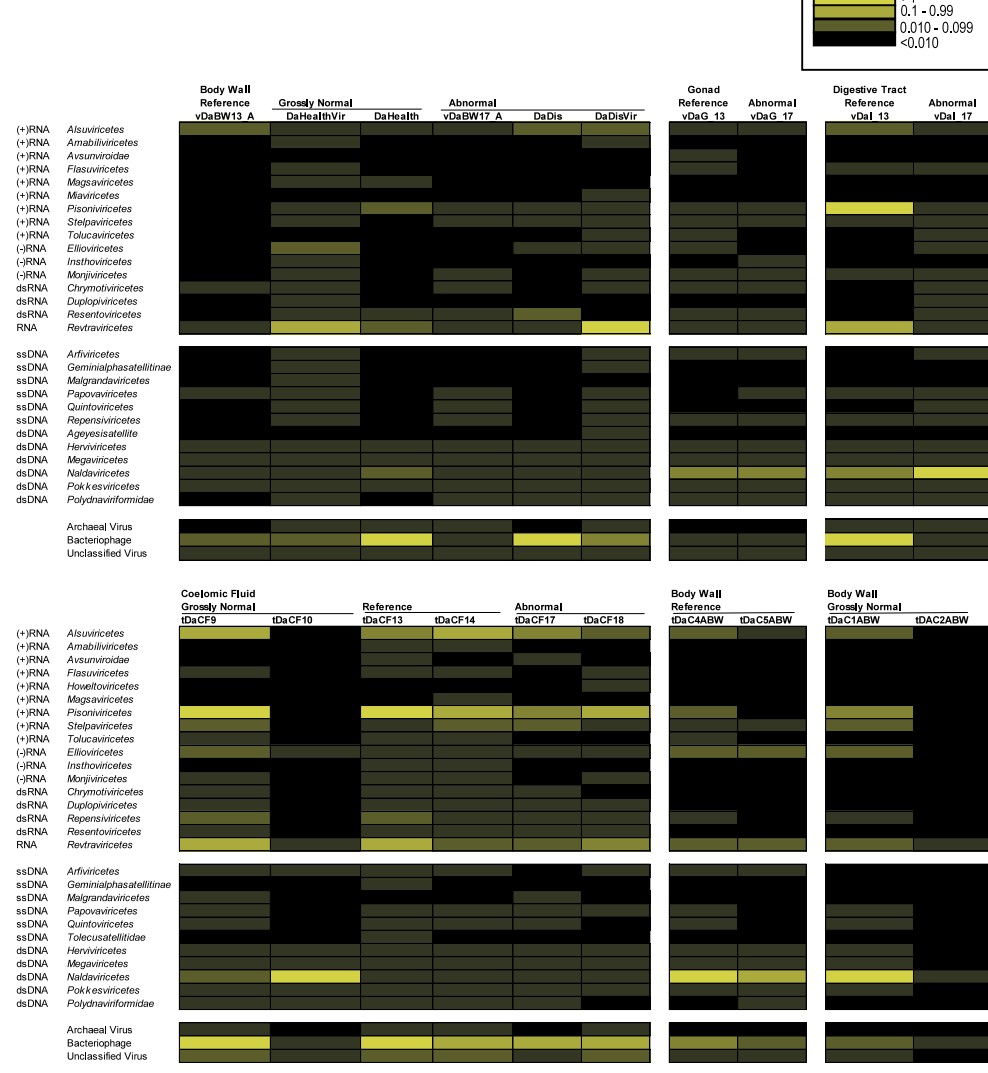

**Figure 1 Heat map representation of viral orders in viral metagenomes (top) and transcriptomes (bottom) prepared from *Diadema antillarum* annotated by Kraken2.** Yellow hues indicate more sequences than grey hues. Black indicates no viruses were detected.

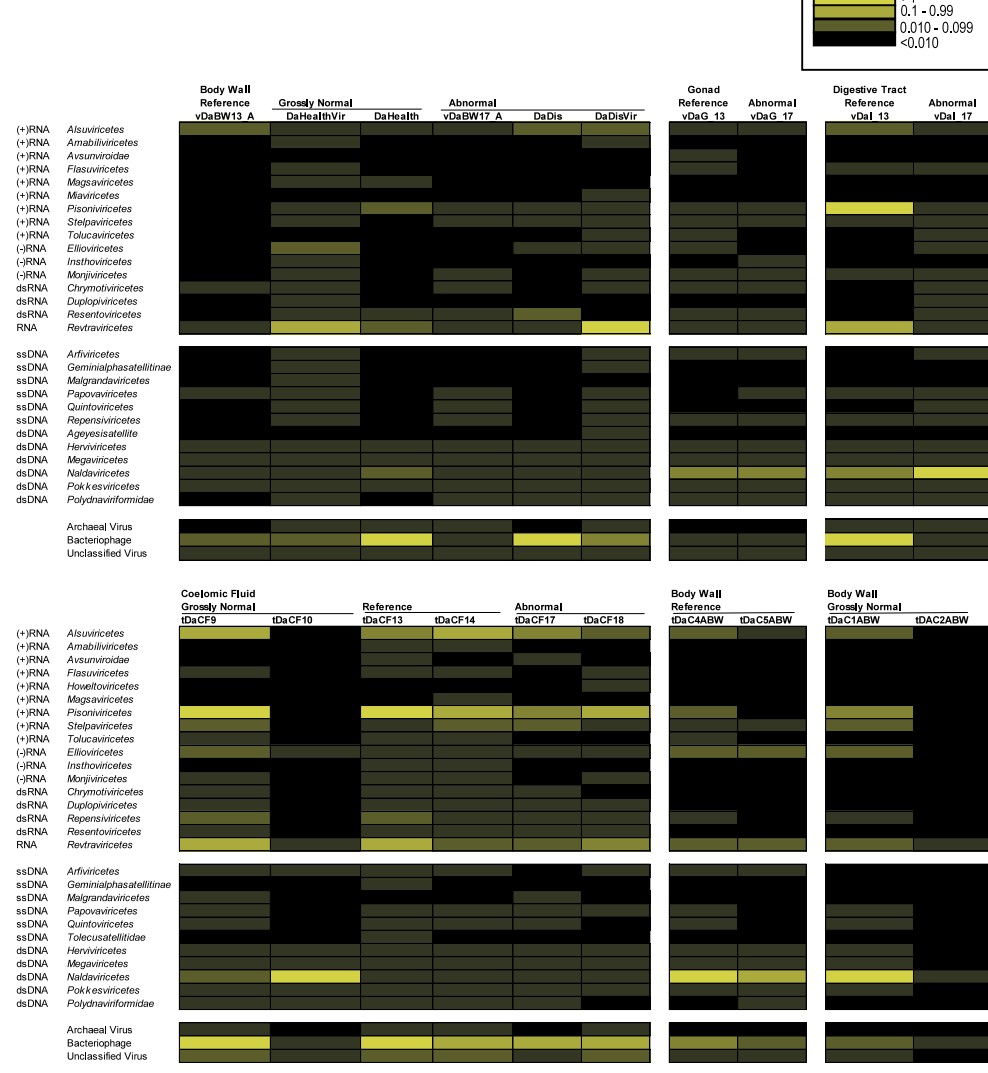

low number of viral reads may have prevented their assembly and representation amongst contig spectra.

### *Holothuria floridana* RNA viral metagenomes

Contig spectra from *H. floridana* were dominated by *Monodnaviria* (*Cressdnaviricota*, *Petitvirales*; 66.8% in individual libraries; 67.2% in global assembly), followed by double stranded DNA bacteriophage (25% in individual libraries; 24.7% in global assembly). The remaining sequences matched one *Mouviricetes* (global assembly), a *Magsaviricetes* (1 each

in FKSCJan and global assembly; Fig. 2), and *Pisoniviricetes* (7.7% of individual and 6.5 in global assembly; Fig. 3).

### Ribosomal RNAs in viral metagenomes

Comparison of contig sequences to the SILVA database (accessed May 2023, release 138.1) revealed that 736 contigs matched eukaryotic and bacterial ribosomal RNAs. These contigs matched several putative sequencing contaminants including *Pseudomonas, Klebsiella,* and *Meiothermus*. Marine bacteria that were well represented in SILVA hits included *Vibrio* spp., *Halarcobacter,* and *Terasakiella* and *Halodesulfovibrio*. Eukaryotic 18S and 28S rRNA genes yielded four contigs from both global and individual assemblies matching the 18S rRNA gene of the DaSc-associated *Philaster* clade, which was an expected constituent of the microbiomes of DaSc-affected urchins, the ciliate *Euplotes*, as well as basidio- and ascomycete fungi (Fig. 4), including *Malassezia restricta*, which may constitute a sequencing contaminant.

### Comparison between viral metagenomes and transcriptomes

Reciprocal BLASTn of all viral contigs amongst *H. floridana* libraries revealed that most ($n = 37$) putative viral genome fragments were shared across individual specimens. All of these common viral fragments were dsDNA bacteriophage ($n = 7$), *Cressdnaviricota* ($n = 3$), and *Petitvirales* ($n = 27$), but not *Pisoniviricetes* nor the *Magsaviricetes* fragments. *Pisoniviricetes* contigs were only recovered from libraries prepared using the Sigma WTA2 chemistry. No viral contig recovered from either *D. antillarum* nor *H. floridana* matched any transcriptome contig recovered from DaSc-affected or grossly normal *D. antillarum* from April 2022 (Hewson et al., 2023).

## DISCUSSION

The overarching goal of the present study was to surveil viruses associated with grossly normal and abnormal *D. antillarum*, in an effort to identify candidate pathogenic agents associated with mass mortality, which preceded conclusion that *D. antillarum* scuticociliatosis (DaSc) was caused by the DaSc Philaster clade ciliate (Hewson et al., 2023; Vilanova-Cuevas et al., 2023). Our results provide several useful observations for future investigation of echinoderm viruses and perhaps in the context of investigating future marine mass mortality events. First, our data suggest that the echinoid *D. antillarum*, whether or not affected by DaSc, and across several tissue types, does not have the same detectability of viruses as *H. floridana* or other echinoderms that have been examined in prior work (Gudenkauf et al., 2014; Hewson et al., 2018a; Hewson et al., 2014; Jackson et al., 2022). Second, our data show that *H. floridana* is inhabited by viral orders that have been observed previously in other holothurian species (Hewson, Johnson & Tibbetts, 2020). Finally, our work suggests that viral metagenome preparation strategy may influence the detection of RNA viruses, and that few viruses detected by viral metagenomics are found in transcriptomes prepared from the same specimens.

We expected to discover prominent and common RNA viruses in *D. antillarum* viral metagenomes based on prior surveys of sea stars (Gudenkauf et al., 2014; Hewson, Aquino

**A**

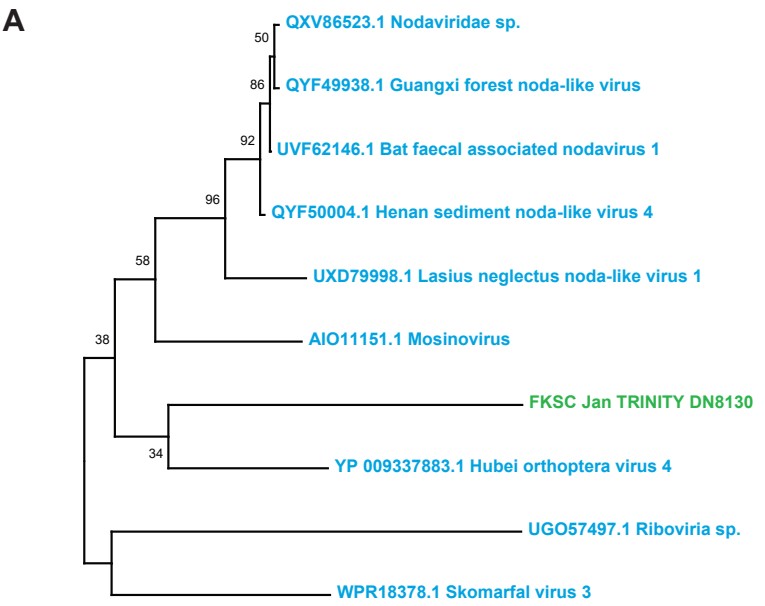

**B**

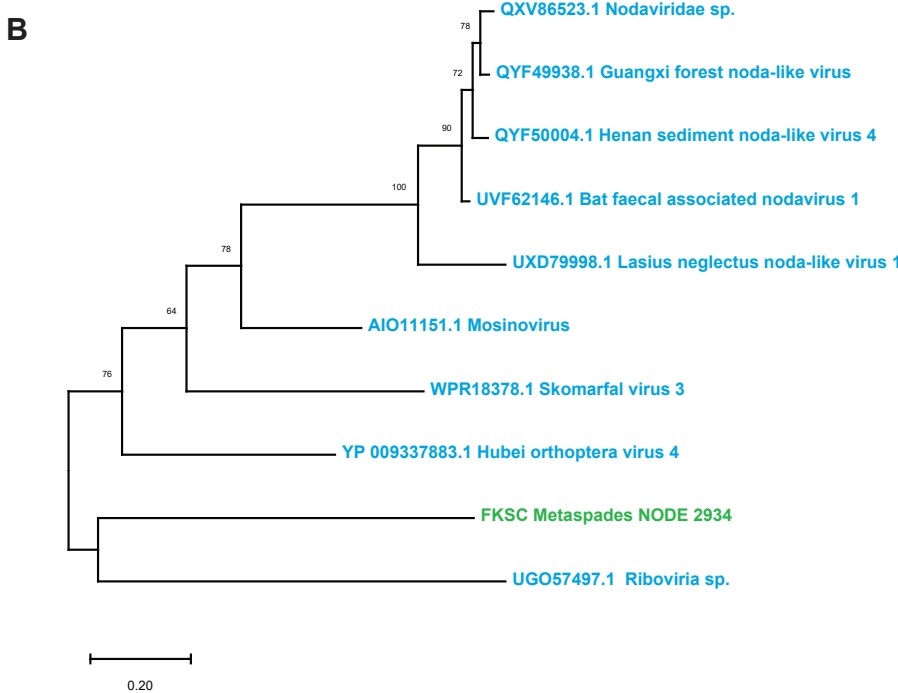

**Figure 2** *Magsaviricetes* **genome fragments recovered from** *H. floridana* **in individual (A) and global (B) assemblies.** Green indicates sequences assembled in this study, blue are closest relatives at NCBI. The reconstruction was performed on a 132 aa and 172 aa alignments (A–B, respectively), using the LG model with gamma distributed sites. Scale bar represents substitutions per site. Numbers above nodes indicate bootstrap values of 1,000 model iterations.

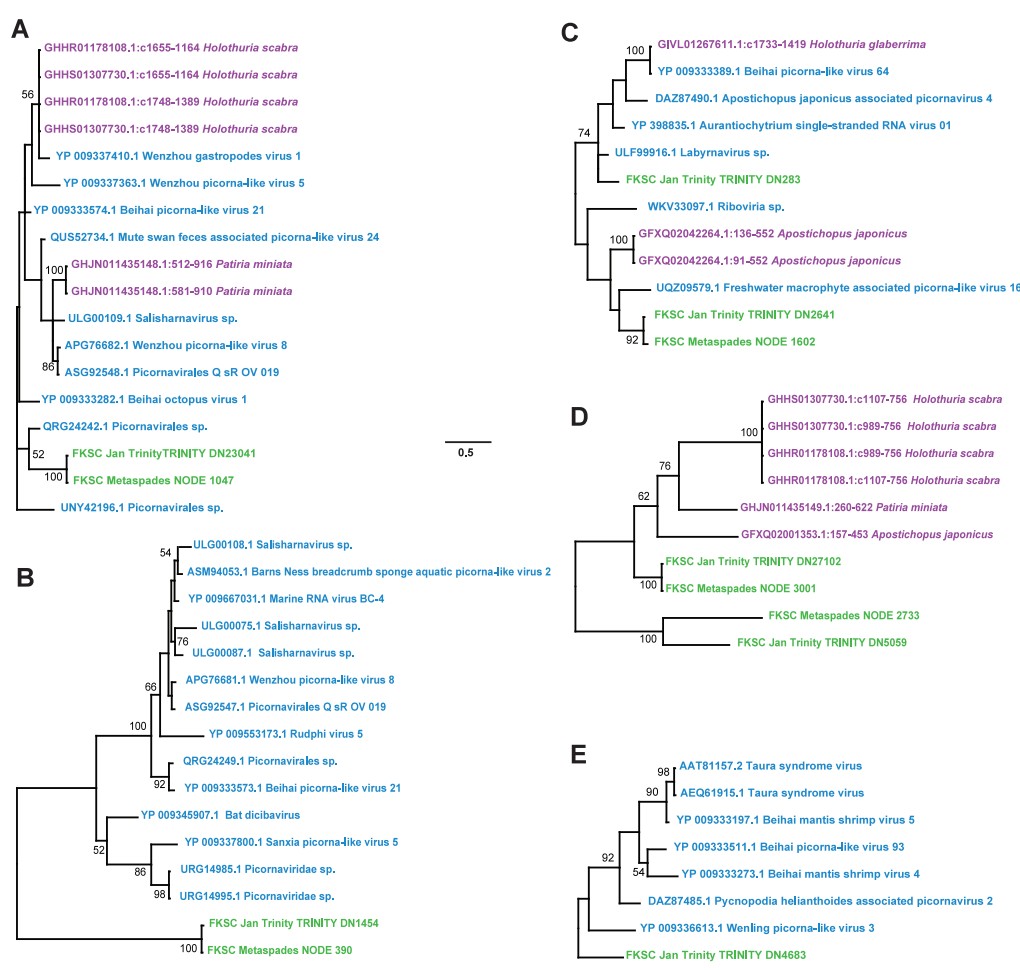

**Figure 3** *Pisoniviricetes* **genome fragments recovered from *Holothuria floridana*.** The phylogenetic reconstructions were performed based on 56 aa, 26 aa, 65 aa, 100 aa, and 128 aa (for A–E, respectively) alignments of overlapping contig regions. Trees were constructed using models reverse transcripts (rtrev) with gamma and invariant sites (A), LG with uniform sites (B), LG with gamma sites (C–E). Green were recovered in this study, blue are viruses in NCBI, and purple are matches to the Transcriptome Shotgun Assembly database (filter 'Echinodermata') at NCBI. Scale bar equals substitutions per site. Numbers above nodes indicate bootstrap values of 1,000 model iterations.

*& De Rito, 2020*; *Hewson et al., 2014*; *Hewson & Sewell, 2021*; *Jackson et al., 2020*), sea cucumbers (*Hewson, Johnson & Tibbetts, 2020*) and those recovered from transcriptomes (*Jackson et al., 2022*). Surprisingly, we did not recover any RNA viral contigs, and only a handful of DNA viral contigs, in both DaSc-affected and grossly normal urchin tissues. While we observed sequencing reads within libraries matching RNA viral genomes, they were less pronounced in viral metagenomes than in transcriptomes. Given that the large number of host sequence matches in read libraries was similar between both *D. antillarum* and *H. floridana*, the lack of viral sequences was not due to poor sequence coverage alone.

Ribosomal RNAs may be copurified with virus particles using the approach employed in this study, which have also been used to identify common protists associated with sea

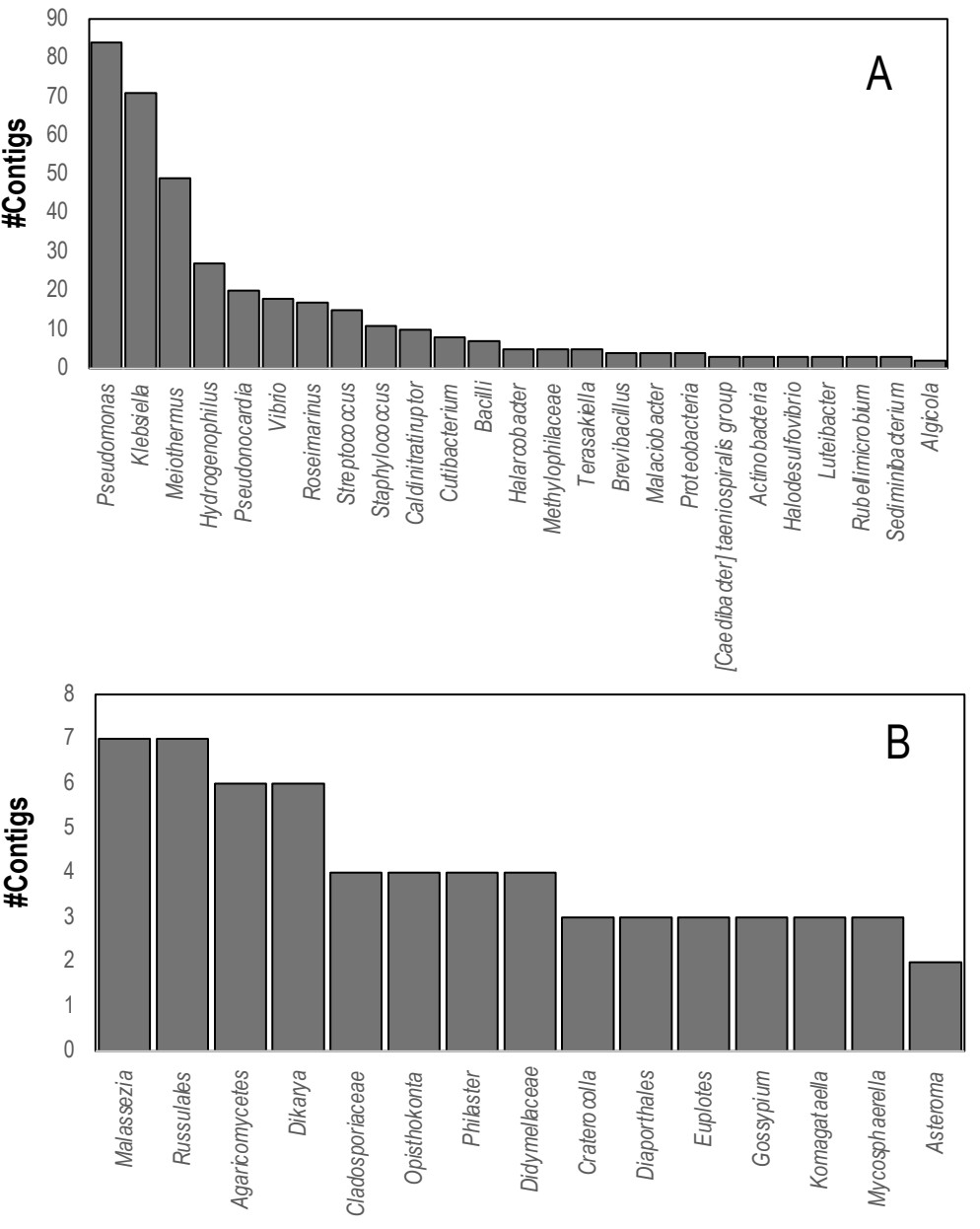

**Figure 4 Contigs matching (A) Bacterial and (B) Eukaryal rRNAs within *D. antillarum* spectra.** Annotation was performed based on BLASTn matches to the SILVA Database using an *E*-value cutoff of $1e^{-100}$.

stars (*Hewson & Sewell, 2021*). We observed marine bacteria, marine ciliate and fungal rRNAs amongst viral metagenomes (Fig. 4). These results further emphasize that the low representation of viruses within *D. antillarum* viral metagenome contig spectra may be partially due to the extensive cellular microbial RNA and DNA present in material used to prepare viral metagenomes, possibly due to coextracted compounds that interfere with nucleases used to prepare viral metagenomes.

In contrast to *D. antillarum* viral metagenomes, *H. floridana* bore viral taxa that have been previously observed in sea cucumbers (*Hewson, Johnson & Tibbetts, 2020*) and more widely in echinoderm viral metagenomes (*Hewson, Aquino & De Rito, 2020*; *Hewson et al., 2018b*; *Jackson et al., 2022*; *Shi et al., 2016*). *Pisoniviricetes* sequences from *H. floridana* were most similar to picornaviral genomes recovered from transcriptome surveys of aquatic invertebrates (*Shi et al., 2016*). When compared to the TSA archive at NCBI, sequences matched at 37–67% nucleotide identity to transcriptome assembled fragments from sea cucumbers (*Holothuria scabra, Holothuria glabberina, Patiria miniata,* and *Apostichopus japonicus*) (Fig. 5; Table S2). Within the *Pisoniviricetes*, one sequence placed closest to *Labyrnavirus*, while others were most similar to viruses recovered from crustacean transcriptomes (Fig. 3). *Pisoniviricetes* have been recovered from a variety of habitats including free in plankton (*Culley, Lang & Suttle, 2003*; *Culley, Lang & Suttle, 2006*; *Moniruzzaman et al., 2017*; *Solomon & Hewson, 2022*), sediments (*Zhang et al., 2024*) and associated with a myriad of invertebrates (*Shi et al., 2016*). Their roles in disease are unclear since they are typically recovered from grossly normal specimens. Comparison of positive-sense ssRNA viruses recovered in this study against assembled shotgun transcriptomes revealed similarly to those recovered from other sea cucumbers, but at least two contigs matched viruses recovered from the chytrid *Schizotrichium* sp. Hence, while most sea cucumber ssRNA viruses recovered in this study likely inhabited echinoderm tissues, some may also infect co-occurring eukaryotic microorganisms that form part of the echinoderm microbiome.

The *Magsaviricetes* sequences recovered from *H. floridana* were most closely related to sequences retrieved from surveys of uncultivated crustacean viruses. The *Magsaviricetes* genome fragments recovered in the *H. floridana* survey fell within a group of viruses discovered through metagenomics, including those from grasshoppers, mosquitoes, and ants, as well as those recovered from environment surveys of estuarine (riverine) sediments and water (Fig. 2). *Magsaviricetes* infecting shrimp have been reported in the sea cucumber *Apostichopus japonicus*, suggesting that they are a secondary host (*Wang et al., 2021b*), and cause significant histopathologic changes in internal organs during infection (*Wang et al., 2021a*). However, as no gross abnormalities were noted in the surveyed specimens, it is unclear whether *Magsaviricetes* in *H. floridana* are infectious.

Double and single-stranded DNA viruses were also detected in this study, which may have resulted from active transcription of viral genes, or because ssDNA is not effectively digested by nucleases used to prepare viral metagenomes. Amongst *Monodnaviria*, Petitvirales (*Malgrandaviricetes*; *Microviridae*) infect bacteria and comprised a large number of contigs. It is worthwhile to note that their large representation in the contig spectra was not a consequence of amplification biases as noted in previous works (*Dunlap et al., 2013*; *Hewson et al., 2013*) since our approach did not employ rolling circle amplification *via* phi29 polymerase. These viruses likely infect microbiome constituents, since all currently described microviruses infect intracellular parasitic bacteria, *Enterobacteriaceae*, and *Spiroplasma* (*Kirchberger & Ochman, 2023*; *Roux et al., 2012*). In addition to *Petitvirales*, *Cressdnaviricota* were recovered from both individual and global assemblies of *H. floridana*. *Cressdnaviricota* have been reported to occur in a variety of metazoan hosts (*Rosario, Duffy*

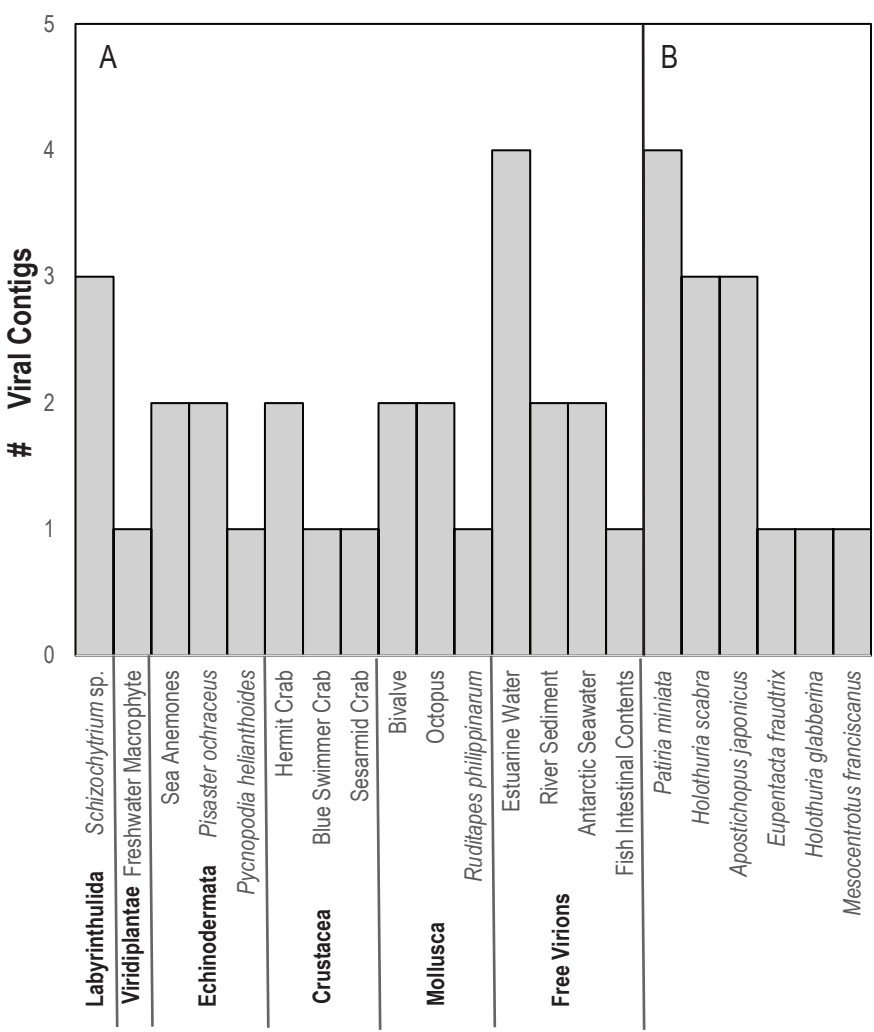

**Figure 5  Number of viral contigs matching (A) viruses in the non-redundant (nr) database and (B) sequences in the transcriptome shotgun assembly (TSA) database at NCBI ordered by host phylum.**

& Breitbart, 2012; Rosario et al., 2015), including echinoderms (Jackson et al., 2016) and may represent metazoan infections. However, to date there remains only scant evidence that any *Cressdnaviricota* cause disease in invertebrates.

While many dsDNA phage sequences ($n = 98$ in *H. floridana* and 5 in *D. antillarum*) were recovered, few could be assigned taxonomically beyond *Caudoviricetes*. In *D. antillarum*, all phage fragments could not be confidently distinguished from putative contaminants in the *Enterobacteriaceae*. In *H. floridana*, bacteriophage sequences matched most closely to those infecting marine hosts. The wide detection of phage sequences both within contig spectra and in individual read libraries may indicate that viruses of microbiome constituents are common, and that their large proportion of sequence space may prevent observation of less abundant metazoan viruses.

The lack of overlap between contigs assembled from viral metagenomes and transcriptomes may indicate that viral particle purification can affect overall viral composition when compared to bulk RNA extracts. Hence, transcriptome mining studies (*e.g.*, *Jackson et al., 2022*) may reflect different viral taxa than those recovered by purifying and sequencing virus particles.

It is unclear why *Pisoniviricetes* (and other ssRNA viruses) were recovered by the Sigma Aldrich TransPlex WTA2 kit across several specimens, but not the Sigma Aldrich SeqPlex kit, since both employ similar chemistries. The Seqplex kit first reverse transcribes RNA using manufacturer-proprietary semi-degenerate 3′- and universal 5′-ends, whereas the TransPlex WTA2 kit uses complementary primers composed of a quasi-random 3′ end and a universal 5′ end. The manufacturer supplied semi-degenerate 3′end may bias towards cellular organisms (or against viruses) through unknown specificity. While amplification with both chemistries was performed for only one *D. antillarum* sample and one *H. floridana* sample (where both SeqPlex amplified libraries yielded no viral contigs), it is also possible that variation in viral detectability was due to different tissues used for other samples that we surveilled in this study. It is recommended that future studies of RNA viruses following this approach carefully evaluate amplification strategy to ensure efficient viral recovery from tissue specimens.

## CONCLUSIONS

This study is the first to report viruses associated with two common coral reef invertebrates, *Diadema antillarum* and *Holothuria floridana*. While we were successful in recovering viral contigs typical of those identified in marine invertebrates from *H. floridana*, very few viral reads were identified in *D. antillarum*. We recommend careful choice of RNA amplification strategy to reduce biases associated with library preparation when considering viral metagenomic surveys of echinoderms. Viral metagenomic survey of viruses may be of utility to coral reef conservation by identification of potential agents affecting animal health, especially when followed by studies examining their pathogenicity. For example, observation of viruses associated with specific disease processes may be isolated and challenged against naïve hosts to satisfy *e.g.*, *Koch (1893)* postulates. Hence, we recommend wide viral metagenomic surveillance of critical coral reef species to discern between viruses associated with normal specimens and those associated with mass mortality events.

## ACKNOWLEDGEMENTS

The authors are grateful to K Kitson-Walters, M Pistor, T Cornwell, T Wijers, M van der Laan, J Antoine, and W Sharp for assistance with collecting urchins and their processing. We thank C Kellogg for comments on an early manuscript draft.

### Funding

This work was supported by NSF OCE- 2049225 and a David R. Atkinson Center for Sustainable Futures rapid response grant awarded to Ian Hewson. The funders had no role in study design, data collection and analysis, decision to publish, or preparation of the manuscript.

### Grant Disclosures

The following grant information was disclosed by the authors:
NSF: OCE- 2049225.
David R. Atkinson Center for Sustainable Futures.

### Competing Interests

Mya Breitbart is an Academic Editor for PeerJ.

### Author Contributions

- Ian Hewson conceived and designed the experiments, performed the experiments, analyzed the data, prepared figures and/or tables, authored or reviewed drafts of the article, and approved the final draft.
- Marilyn Brandt performed the experiments, prepared figures and/or tables, authored or reviewed drafts of the article, and approved the final draft.
- Kayla Budd performed the experiments, prepared figures and/or tables, authored or reviewed drafts of the article, and approved the final draft.
- Mya Breitbart performed the experiments, prepared figures and/or tables, authored or reviewed drafts of the article, and approved the final draft.
- Christopher DeRito performed the experiments, analyzed the data, prepared figures and/or tables, authored or reviewed drafts of the article, and approved the final draft.
- Samuel Gittens Jr performed the experiments, prepared figures and/or tables, authored or reviewed drafts of the article, and approved the final draft.
- Michael W. Henson performed the experiments, analyzed the data, prepared figures and/or tables, authored or reviewed drafts of the article, and approved the final draft.
- Alwin Hylkema performed the experiments, prepared figures and/or tables, authored or reviewed drafts of the article, and approved the final draft.
- Moriah Sevier performed the experiments, prepared figures and/or tables, authored or reviewed drafts of the article, and approved the final draft.
- Matthew Souza performed the experiments, prepared figures and/or tables, authored or reviewed drafts of the article, and approved the final draft.
- Brayan Vilanova-Cuevas performed the experiments, analyzed the data, prepared figures and/or tables, authored or reviewed drafts of the article, and approved the final draft.
- Sarah Von Hoene performed the experiments, prepared figures and/or tables, authored or reviewed drafts of the article, and approved the final draft.

## Field Study Permissions

The following information was supplied relating to field study approvals (i.e., approving body and any reference numbers):

D. antillarum were collected from St. Thomas under permit DFW22033U and St. John under permit VICR-2022-SCI-0007, and in Saba under auspices of the Saba Conservation Foundation, management authority of the Saba Marine Park. Specimens of Holothuria floridana were collected under permit FKNMS-2023-057 issued to the Florida Fish and Wildlife Conservation Commission (W. Sharp).

## Data Availability

The sequences produced in this study are available at NCBI: PRJNA1117494 and SRR29258987–SRR29259000. The viral contigs are available at NCBI: PP872373–PP872393.

The raw sequence data is available at figshare: Hewson, Ian (2024). *Diadema antillarum* and *Holothuria floridana* raw sequence reads. figshare. Dataset. https://doi.org/10.6084/m9.figshare.26042650.v1.

## Supplemental Information

Supplemental information for this article can be found online at http://dx.doi.org/10.7717/peerj.18321#supplemental-information.

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
