# Peer review of "Viral metagenomic investigation of two Caribbean echinoderms, Diadema antillarum (Echinoidea) and Holothuria floridana (Holothuria)"

_PeerJ, doi:10.7717/peerj.18321_

## Round 0.1 · original submission · Major Revisions

Although reviewers #1,2 have given minor revision, reviewer # 3 has given different opinion on your work. The major comment of reviewer #3 is throughout the manuscript, there is no support or explanations for particular assessments. For example, the authors only examined viruses from four groups instead of testing their data against all viral genomes within NCBI and did not explain why they excluded all other viruses. The experimental design is questionable. There are no replicates for the third site mentioned, and the tissue types are not replicated within sites or between sites. In your revision or redo the work, you need to provide convincing answers for all reviewers especially reviewer #3. The paper will be subjected to the acceptance based on your revision and responses especially for reviewer #3.

Reviewer 1 ·

Basic reporting

no comment

Experimental design

no comment

Validity of the findings

no comment

Additional comments

Title: Needs to be changed. Please provide a clear and concise title.

Introduction: Acceptable.

Materials and Methods: Requires extensive revision. Indicate whether references are available for the methodology or if it was developed de novo. If no references are cited, provide a rationale for adopting and following this specific methodology.

Results and Discussion: Requires improvement. Results should be presented first, with significant values clearly indicated. Proper use of statistical tools is necessary. The discussion should flow continuously; consider removing subheadings for better coherence.

Conclusion: Requires rewriting. Use direct and concise sentences to convey the key points clearly.

Annotated reviews are not available for download in order to protect the identity of reviewers who chose to remain anonymous.

·

Basic reporting

The article is clear and easy to follow. The science is sound and justified throughout the manuscript following the scientific method 100%. All references are cited appropriately to justify statements made in the manuscript. References are up to date, as well as relevant to each statement throughout the manuscript.

Figures are all appropriate and professional in appearance.

Experimental design

Research design correlates well to the research question, which adds to scientific knowledge needed in this field of study. Current lack of understanding of viruses in relation to marine invertebrates currently needs more work, and this study adds information to the field of study. This study also highlighted difficulties in genetic analyses, hence future research is needed. Methods highlight a rigorous approach to answer the research question. All permits were identified.

Validity of the findings

Findings were valid and justifiable. Conclusions identified a need for more research in this area of study.

Additional comments

The following are minor comments for Review of Manuscript #102241
INTRODUCTION
Line 48: The previous line mentions echinoderms, however in this line you mention echinoids, which are one group of echinoids. Recommend you transition into echinoids. Such as, “In coral reef ecosystems, echinoids which are one group of echinoderms contribute…”
Line 50: Recommend…Another group of echinoderms, the holothurians…for better flow.
Lines:47-56: Any recent citations to be added 2020s?
Line 58: simplify “crucial constituent”
Line 60: Should add citations from Peter Glynn 1984
Line 73: reword/remove “is a common sea cucumber” since you already mentioned it before
Line 78 “Its fecal pellets may persist for longer” ADD “period” if you meant duration in time
Line 81 assessed by who? Citation?
Line 103: add U.S. Virgin Islands
Lines 105-110 Move to Results section

MATERIALS & METHODS
How many sea cucumbers were collected per site?
ADD Site longitude and latitudes
Please ADD serial numbers for any kits used
Line 188: MEGA X

RESULTS AND DISCUSSION
Line 202: what other holothurian? Provide examples.
Figures 3 and 4, what do numbers above branches represent? Bootstrap values? Maximum likelihood? Maximum parsimony?

Reviewer 3 ·

Basic reporting

The manuscript lacks a clearly defined research question, including an explanation for investigating both species. An explanation should be provided as to why the authors are looking at healthy vs. diseased viromes from the urchin species (D. antillarum) in two locations and a healthy sea cucumber species (H. floridana) in Florida (a different location than the urchin species). It is unclear how these two different investigations are connected and why they are being explored together. Providing more information on how the investigations are connected and clear research questions may help with this.

Additionally, the figures and naming conventions are not clear and hard to discern. In Figure 1 and Table 1, the Library names provide no information, these should be relabeled to convey the sample type. More information should be provided in the captions for all of the figures to explain the samples being examined and what is occurring. Figure 1 should also include a scale bar of color so the audience knows the numerical values of the yellow color.

Experimental design

There is a fundamental flaw in this study with the lack of sample replication. After examining Table 1, for D. antillarum, there are:
2 replicates from body walls of healthy individuals at site STT
2 replicates from body walls of diseased individuals at site STT

1 sample from the body wall of a healthy individual at site STJ
1 sample from the body wall of a diseased individual at site STJ
1 sample from the digestive tract of a healthy individual at site STJ
1 sample from the digestive of a diseased individual at site STJ
1 sample from the Gonad of a healthy individual at site STJ
1 sample from the Gonad of a diseased individual at site STJ

There are no replicates for the third site mentioned in the introduction or methods, and the tissue types are not replicated within sites or between sites. There is mention of using past transcriptomic data for D. antillarum (Supplemental Table 1), however, there is again a replication problem where there are two samples from the body wall from healthy individuals (from the third site Saba), with no diseased samples being offered at this site. There are also two replicates of a different tissue type (Coelomic Fluid) at the third site, which is not a tissue type collected from the other sites. Therefore, comparisons between or across locations or tissue types cannot be made. This investigation does not meet the technical standards for publication.

Additionally, for the RNA preparation comparison made, this part of the study also lacks replication. Only one healthy sample and one diseased sample from D. antillarum at STT were assessed using the TransPlex WTA2 kit along with one sample from H. floridana, which is not enough samples to make claims on RNA preparation approaches between the TransPlex WTA2 kit and the SeqPlex kit. There should be replication of the RNA preparation methods within the same types of samples to make those comparisons and statements.

The only part of this study that has sufficient replication is the H. floridana samples, which have 4 replicates. The authors could rework this manuscript to include only those samples.

Additionally, there is not enough detail in the methods for this to be replicated by others. Parameters used in the bioinformatics pipeline should be added, along with reasoning for why they were selected.

Validity of the findings

Throughout the manuscript, there is no support or explanations for particular assessments. For example, the authors only examined viruses from four groups (Line 167) instead of testing their data against all viral genomes within NCBI and did not explain why they excluded all other viruses. Other examples include not providing an explanation of their phylogenetic tree building (what parameters were used and why) and their methodology for identifying viral contigs. Additionally, the methods used in this investigation do not satisfy the field’s standard of capturing viral information or removing host information from the data being analyzed; because of this, along with the replication issue, what we see in their data may just be due to contamination. The authors should include more replicates and improve their pipeline to remove host information (including multiple alignments to host genomes to be certain all host information is removed) and annotate viral contigs in a more robust manner than BLASTn. Additionally, phylogenetic trees should be constructed in a more robust manner than in MEGAX, where you can control the parameters and report them. A standard option would be to use the mafft aligner with IQTree to generate trees.

---

## Round 0.2 · Major Revisions

According to 2 reviewers, this manuscript needs a further revision. Authors have made adequate first revision, however still the revision is needed based on the reviewers comments.

Reviewer 1 ·

Basic reporting

-

Experimental design

-

Validity of the findings

-

Additional comments

Abstract:
Please mention any significant data points that underline the importance of the study.
Introduction:
Please explain how these viruses might affect holothurian health.
Material & method
Please provide a table summarizing the latitude, sampling method, season, and environmental variables (such as temperature and salinity) for each sampling location.
Result:
Please link the data directly to your research questions or hypotheses.
Conclusion:
Emphasize how understanding viral diversity in these species can inform efforts to protect coral reef ecosystems.
Suggest practical applications or future research directions.

·

Basic reporting

The author (s) have fulfilled all recommended changes requested by the reviewers, and have added all supporting citations with regards to any recommended changes.

Experimental design

Better flow and much clearer with an identified research question.

Validity of the findings

Findings are valid

Reviewer 3 ·

Basic reporting

I thank the authors for the adjustments they made to this manuscript. The adjustments have led to a clear and defined focus of this work. They have provided explanations of the goals of this work and why they chose the specific study groups, which has strengthened this manuscript. I have a few comments below about the structure of the article (the results section and figures 2-4). Briefly, I think it would be beneficial to add Figures 2-4 to the results section. Additionally, some of the content in the Discussion sounds like it may fit better in the result section.

Experimental design

The authors have made adequate adjustments to my previous comments and have added more detail describing the bioinformatic methods. Additionally, the authors clearly defined the scope of the work in their adjustments to the manuscript which helped in the understanding of their experimental design. However, I do have suggestions the authors should make to strengthen the manuscript, see below for the detailed suggestions.

Validity of the findings

I still believe there should be more replication, but since the authors have made it clear that this is purely a survey, what they have is sufficient. They have made adequate adjustments in this area as far as adjusting their conclusions and the presentation of results and discussion. I have made comments below for the authors to clearly state the number of replicates examined in particular places in the manuscript, along with clearly identifying what was past work.

Additional comments

Detailed Comments:
Line 28: typo – there is an extra ‘of’ in the line

Line 146-148: Please add how many samples were extracted with both kits (n=x)

Line 192-193: Please add how many samples/replicates were compared in each group (i.e. grossly normal and DaSC-affected): (n=x). Also, please describe how these comparisons were made. Did you use Bowtie, another program, or a custom script?

Line 223-224: Please remove this sentence or reword it “The majority …” Looking at Figure 1 it doesn’t look like that statement is the case. Also please see the next note, which may help with this.

Figure 1: Please add a numerical scale indicating the values associated with the colors for the heatmaps (A max to zero scale is fine, but you need something to indicate the values associated with the colors).

Figure 1: Please indicate what is newly sequenced from this study and what is from past studies; maybe add an asterisk next to previously sequenced samples as an example.

Figures 2-4: These figures are not found in the result section—I’m not sure if that was a mistake or not; they look like they should be in the result section since they are not summary figures. Because of this, the first paragraph in the discussion section seems a little disjointed; the authors state the main takeaways based on the results that the reader hasn’t encountered yet. The first paragraph in the discussion is a nice paragraph; I think it would be beneficial for the reader to see some of the results before they get there.

Lines 273-286: This paragraph sounds like it should be in the Results section.

Tables 2-4: These blast output tables should be in the supplement. It may be more beneficial to the reader to distill this into a figure. I could imagine something like host taxa on the x-axis and counts of viruses on the y-axis as stacked bar plots; that may be more useful to a reader.

Line 301: this is an awkward sentence (the” similarly mainly” part), please re-work

---

## Round 0.3 · accepted · Accept

Thank you for submitting your revised manuscript. After reviewing the changes made by authors, I am pleased with the improvements made, and I am happy to recommend "ACCEPT" your paper for publication. You have adequately addressed the previous concerns and the paper's content is now clear and scientifically robust.